# From Cyberpunk to Cramped Dweller: The Peculiar History of Hong Kong 'Heterotopias'

Daniel McCoy 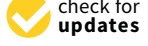

Department of History, Northern Illinois University, DeKalb, IL 60115, USA; Z1856323@students.niu.edu

**Abstract:** 75.6% of land comprising Hong Kong remains undeveloped according to the special administrative region's planning department. In turn, Hong Kong's constricted real estate, now estimated to be the world's costliest, has created eye-popping living arrangements historically and contemporarily. Denizens' colorful reputation and imagination for flouting city ordinances, zoning laws, and spatial management stand emblematic of tenacious self-sufficiency and a free-spirited brand of runaway capitalist initiative. Why is this conspicuous trademark of Hong Kong's societal fabric very much alive in the 21st Century? Why does this matter in a rapidly urbanizing world witnessing the ascension of mega-urban centers alongside ever-widening socioeconomic chasms? This paper intends to illuminate the peculiar origins and longevity of the Kowloon Walled City, an urban monolith of notoriety and autonomy that blossomed in a semi-legal grey zone unencumbered under British protectorate rule for nearly a century. Parallels will connect the linear trajectory between Kowloon's hardnosed living to today's comparable Chungking Mansions and the hundreds of thousands of cage homes appearing in all corners of the city. This paper aims to answer why these residential paradoxes continue to function with efficiency and relevancy, posing solutions for indigent housing while exacerbating the stigma of social and economic ostracism.

**Keywords:** Hong Kong; housing; Kowloon Walled City; Chungking Mansions; cage homes; squatter settlements; Great Britain; demographics

## 1. Introduction

Hong Kong's 2019 summer of discontent transitioned into an autumn of pervasive, combustible rage [1]. Initially inspired by the threat of an extradition bill designed to send Hong Kongese facing criminal offenses to mainland China for trial and sentencing—a nation with a dubious human rights record at best—protesters in the beleaguered former British colony maintained their opposition to Beijing creeping into Hong Kong's affairs. The Hong Kong government, under the leadership of Chief Executive Carrie Lam Cheng Yuet-ngor, threatened to legislate the extradition bill. The public viewed this action as an egregious violation of the political, social, and economic autonomy that had been granted to Hong Kong as a Special Administrative Region (SAR) of China under the Sino-British Joint Declaration on the Question of Hong Kong. The concept of Hong Kong as a SAR roots itself in two crucial moments: 25th Hong Kong Governor Murray MacLehose's visit to Beijing in 1979 and Margaret Thatcher and Deng Xiaoping's summit meeting in 1982 commencing Sino-British negotiations over Hong Kong's future sovereignty [2] (p. 2). The SAR became the product of a bilateral agreement signed between the People's Republic of China (PRC) and Great Britain in 1984 and actualized by the Handover of 1997. Under the "one country, two systems" plan outlined in the 1984 agreement, beginning on July 1, 1997, Hong Kong's liberalized social and economic systems would remain free of interference from Beijing's political orbit for the next fifty years. The 2019 protests in Hong Kong were pushback to Beijing's lack of concealment in its accelerated erosion of Hong Kong's coveted SAR status and a rebuke of the Hong Kong Government's erosion of its constitutional statue known as the Basic Law of the Hong Kong Special Administrative Region of the People's Republic of China.

While the extradition bill died on the vine, the protestors' political and social platforms of grievances evolved. One such grievance concerned the multilayered problem of socioeconomic inequality evidenced by the lack of affordable housing alongside governmental mismanagement of land, two long-standing problems since the earliest days of the former colony's British patronage. The issue of inadequate public housing in one of the wealthiest locales on Earth tethers itself to an urban history long rife with paternalistic policymaking defined by crammed tenements, squatter settlements, class tensions, and ideological clashes along civilizational lines. These factors nagged a sizeable demographic, and the scars of which appear in the present-day SAR. As a result, the chasm between Hong Kong's haves and have-nots steadily widens. Despite its contemporary and historical disparities stemming from British paternalism, Sinophobia, and social inequality, Hong Kong's peculiar housing arrangements also personify a cornerstone of Hong Kong's outward image and metropolitan trajectory. The SAR's public housing system is the life source for those who call these notable structures home.

Approximately seventy-five percent of land comprising Hong Kong remains undeveloped according to the SAR's planning department. "Hong Kong is not short of land", stated third Chief Executive Leung Chun-ying in 2012, despite "limited resources in public finance" being a prevailing hindrance in the SAR's urban development [3] (p. 25). Resultantly, Hong Kong's constricted real estate, now estimated to be the world's costliest, has created eye-popping living arrangements both historically and contemporarily. Denizens' and the local government's reputation and imagination for flouting city ordinances, zoning laws, regulations, and spatial management stand emblematic of tenacious self-sufficiency and a brand of runaway capitalist initiative. Why is this conspicuous trademark of Hong Kong's societal fabric very much alive in the twenty-first century? Why does this matter in a rapidly urbanizing world witnessing the ascension of mega-urban centers alongside ever-widening socioeconomic chasms? What explains the longevity and relevancy of long-known residential dark corners of the former Colony or "heterotopias" as philosopher Michel Foucault coined [4] (p. 24)? Foucault conceptualized the bipolar functionality of a heterotopia "to create a space of illusion that exposes every real space, all the sites inside of which human life is partitioned, as still more illusory . . . [or] else, on the contrary, their role is to create a space that is other, another real space, as perfect, as meticulous, as well arranged as ours is messy, ill constructed, and jumbled [4] (p. 27)".

This paper intends to illuminate the peculiar origins and longevity of the Kowloon Walled City (九龍城寨), an urban monolith of notoriety and autonomy that blossomed in a semi-legal grey zone unencumbered by British protectorate for nearly a century. Parallels will connect the linear trajectory between Kowloon's hardnosed living to today's comparable Chungking Mansions (重慶大廈) and the hundreds upon thousands of cage homes appearing in all corners of the city. Additionally, municipal responses and actions, both progressive and regressive, to the prevalence and demise of squatter settlements thread the narrative throughout. This paper does not, however, examine comparable housing on mainland China, nor does it offer a comparative analysis between the mainland and Hong Kong's housing policies and housing histories. This paper aims to answer why Hong Kong's residential paradoxes continue to function with efficiency and relevancy, posing solutions for indigent housing while exacerbating the stigma of social and economic ostracism.

The primary and secondary source material detailing examples and analysis of Hong Kong housing in this paper encompasses an interdisciplinary approach, derived from a selection of anthropology, sociology, and socioeconomic studies. In turn, this investigation uncovers a noticeable lack of pre-existing historical analysis which it intends to rectify. This paper, through a critical lens of history, aims to tether much of the Hong Kong's modern strife to deeper historical roots representative of the effects of colonial governance, fundamentally flawed urban planning, and citizen-led self-determination.

With an area measuring slightly over 400 square miles, the Hong Kong Special Administrative Region of the People's Republic of China consists of four major areas [5]

(p. 5). Hundreds of outlying islands dot Hong Kong's waterways, while Hong Kong Island flaunts its financial center and historical buildings [5] (p. 5). Kowloon, meanwhile, contains densely populated areas as well as the teeming tourist epicenter of the SAR [5] (p. 5). Finally, the New Territories stretch to the border with mainland China and encompass half of Hong Kong's population [5] (p. 5). In 2005, *The Economist* named Hong Kong the "freest economy in the world", occupying the cutting-edge of global neoliberalism [6] (p. 178). Following 156 years of British rule, Hong Kong was reabsorbed into Chinese sovereignty on 1 July 1997, during the ceremonial handover between Great Britain and Communist China. Despite officially being part of China, specifically in matters of foreign relations and defense, Hong Kong would reap a tremendous degree of political, social, and economic autonomy as granted by the Handover for the next fifty years after 1997. This streak of independence fosters an image the world has come to harbor of Hong Kong as a miniscule locality projecting world-renowned distinction and individuality.

Hong Kong features the second largest public housing system in the world after Singapore. While the total landmass of Hong Kong stands at 1104 square kilometers, most of the SAR's population resides on 120 square kilometers factoring into 58,000 people per square kilometer [7] (p. 1391). In *Planet of Slums*, urban historian Mike Davis writes that in contrast to Europe, whose "saucer" cities reveal the indigent clustered in high-rise housing on the urban periphery, "public housing for the poor in the [global] South is an exception [Hong Kong] rather than the rule [8] (p. 31)". In the SAR, Davis notes, "between one fifth and one third of the urban poor live within or close to the urban core, mainly in older rental multifamily housing [8] (p. 31)". While Hong Kong has successfully absorbed millions of refugees and economic migrants from mainland China since the end of the Second World War and incorporated them into the post-war economic miracle that the former British Crown Colony undoubtedly represents, Davis asserts Hong Kong's "success in rehousing squatters, tenement-dwellers, and civil war refugees in new public apartment blocks is not quite the humanitarian miracle often depicted [8] (p. 63)".

## 2. British Hong Kong vs. Chinese Hong Kong

Across the British empire, its cities fostered burgeoning native-population sectors, or "illegal spaces" as termed by anthropologist Alan Smart [9] (p. 21). These "spaces" were segregated and ostracized from the European sections of town yet blossomed with colorful cultures and underground economies [9] (p. 21). These bustling metropolises also exhibited the most glaring characteristic signifying the dichotomy between the West and East and the colonizing haves versus the colonized have-nots: housing. Furthermore, perceived differences in climate adaptability between Europeans and the Chinese materialized into anti-Chinese legislation at the turn of the twentieth century. In 1888, the European District Reservation Ordinance blocked the Chinese from constructing tenements within designated sections of the growing city. An adjoining preamble proclaimed:

> The health and comfort of Europeans in a tropical climate demand conditions which are inconsistent with the neighborhood of houses crowded with occupants and otherwise used after the manner customary with the Chinese inhabitants, and whereas the influx of Chinese into the colony tends constantly to narrow the area of the City of Victoria where such conditions are attainable, and it is desirable to reserve by law a district wherein such conditions may be secured [10] (p. 95).

Historian Ming K. Chan notes that while the imperial decree did not explicitly "discriminate" against Chinese people, it did segregate them to "dwellings they were [only] accustomed to building and inhabiting [10] (p. 95)".

The European contingent in Hong Kong came to view their Chinese counterparts as "irrational, insanitary, and uncivilized [9] (p. 31)". Cramped in their cacophonous, native quarter of Tai Ping Shan (太平山), the indigent Chinese epitomized "the literal embodiment ... of darkness, noise, disease and disorder [9] (p. 31)". The third plague pandemic which crippled Hong Kong in 1895 emboldened British advocacy for segregated

housing between the Europeans and Chinese as well as "sanitary interventions" as part of the British civilizing mission to rear the Chinese in the methods of British-instilled urban decorum and public order [9] (p. 31). In 1902, the Colonial Office allowed for the "reservation" of an area of New Kowloon known as the Peak District "not to be confined to Europeans", thus not excluding "clean-living Chinese of good standing [10] (p. 98)". Smart notes during this time, however, Hong Kong's architecture did not account for the split between the Chinese and British urban spheres because the Chinese primarily lived in structures considered to be of scant "Chinese" cultural contextualization [9] (p. 31). Specifically, the European and Chinese sections of Hong Kong were distinguished by "who lived in the structures and how they lived there" as the Chinese section of the city defined itself by its evident overcrowding and squalor as perceived by Europeans [9] (p. 32).

The British hoisted the Union Jack atop Possession Point on Hong Kong Island on January 26, 1841, effecting the contentious agreement made between the British and Chinese at the unratified Convention of Chuenpi (穿鼻草約). Chinese day laborers tasked to build the new settlements of British Hong Kong predominately resided in huts in the areas of Tai Ping Shan, Tsim Sha Tsui (尖沙咀), and Sai Ying Pun (西營盤) scattered across Hong Kong Island and Kowloon [11] (pp. 87, 91). By 1856, five "premises" occupied Sai Ying Pun on the Village Crown Rent Roll [11] (p. 92). In the following year, the Squatter Board issued thirteen squatter permits for Sai Ying Pun, in particular. By 1858, the number of squatter permits had risen to forty-one. In 1861, forty-six squatters as a group paid "$79.62 for house rent and $11.58 for cultivated land [11] (p. 92)". Apart from four "licenses", the squatters' plots of land in Sai Ying Pun had been sold at Hong Kong Crown government land sales [11] (p. 92). Those who inhabited Sai Ying Pun consisted mainly of "small shopkeepers, craftsmen and labourers [11] (p. 96)". A December 1881 article in *The China Mail* highlighted "the tendency shown by the local Government to compensate native squatters on Crown land at the [expense] of the purchasers of such land [12]". Moreover, the article observed the "'special condition of sale' to which Crown lessees had to agree and under which such compensation had to be arranged with the said squatters, subject to the approval of the Governor [12]". Eventually, the territory comprising Sai Ying Pun, in particular, was relinquished to the Hong Kong Government in 1930 at a sum of HKD 900,000 [11] (p. 97). By the 1930s, court-ordered demolitions of huts were commonplace as were the threat of fines or jail time for trespassing onto property held by the Crown. An article from *The China Mail* dated 1932 captured one Mr. Schofield in Central Police Court unabashedly commenting, "If you persist in squatting there, I give you warning now, that this Summer you may be buried under the earth and die like the people did last year [13]". The erasure of native Chinese squatter settlements in Sai Ying Pun marked another example of the Hong Kong colonial regime doling out land to the native poor only later to subsume it, expelling perceived undesirables from precious land of which the government sought to increase the value.

Before the lease of Hong Kong's New Territories to Britain took effect in July 1898, vibrant Sham Shui Po (深水), positioned directly across the dividing line from Old Kowloon, proved to be a seedy stomping ground for Hong Kong's denizens including gamblers, criminals, and smugglers. Sham Shui Po's strategic position became a nuisance in the British imperial litigation system as criminals could quickly slink back into the haven after running afoul inside British jurisdiction [11] (p. 174). A police inspector remarked in 1875 that Sham Shui Po embodied "a notorious place for old offenders. There are ten gambling houses in the village, and there are twenty old offenders there . . . All the robberies committed in British Kowloon were by these old offenders and their associates gathered there [11] (pp. 174–175)". An article published in the *Hong Kong Telegraph* in 1890 elaborated on the "evils" surrounding gambling in Sham Shui Po: "for crowds of a certain section of the community, including women and even children, to go over in steam launches to Sham-sui-po and pass their Sunday afternoons in filthy Chinese dens gambling at poo-chi [蒲志] and fan-tan [番攤], teaching a new generation the very worst forms of vice [11] (p. 175)". Concurrently, Chinese officials consistently balked at implementing

reforms on these "resorts" of perceived ill repute despite China having already officially outlawed gambling [11] (p. 175). The shrewd British demographic and most of the Chinese citizenry in Hong Kong, alike, got their wish albeit only temporarily when a fire in February 1891 wiped out much of gambling dens [11] (p. 175). Always in lockstep with the evolution of Hong Kong housing, urban fires twofold extinguished and revealed sociopolitical and economic tensions within the Colony time and again.

### 3. Kowloon Walled City

To the north of Hong Kong Island, on the northeastern part of Kowloon peninsula, laid the Kowloon Walled City. First fortified in 1668, the Walled City remained insignificant in Chinese civil and military affairs until the British first occupied Hong Kong Island in 1841 during the First Opium War between Great Britain and the Qing Dynasty of imperial China [14] (pp. 30–31). In 1846, the Viceroy of the Two Guangs (兩廣總督) suggested the construction of a "walled-city", completed the following year [14] (p. 31). The dimensions of the wall measured 700 feet by 400 feet with an average height of 13 feet enclosing an area of 6.5 acres [14,15] (p. 31; p. 62). British officials were impressed by its design. Upon its completion, the Kowloon Walled City transformed into a prominent garrison town, symbolizing a Chinese-constructed bulwark against the perceived Western barbarians' decadence and materialism [14] (p. 32). The fortification boasted "six watch-towers (then occupied as family dwellings), four gateways, a granite parapet with 119 embrasures, and dozens of cannons [15] (p. 62)". The fortress was additionally intimidating through its moratorium of commercial shops inside the Kowloon Walled City until 1898 when China leased to Britain a swath of underdeveloped land that came to be known as the New Territories comprising the mainland area adjacent to Kowloon [14] (p. 33). Before British forces commandeered the Walled City in 1899, the first invaders of the fortification had been Chinese anti-Dynastic Triads and Hakka stoneworkers who extorted and robbed the local population [14] (p. 34). Concurrently, in the late nineteenth century, the Chinese in British-administered Hong Kong looked to the Chinese presence across the harbor for protection, authority, and "patronage", all of which rapidly eroding at the sunset of the Victorian era [14] (p. 35).

Kowloon Walled City symbolized "a donut hole of Chinese sovereignty" in British-held Hong Kong and the New Territories [16]. In the 1898 Convention for the Extension of Hong Kong Territory (中英展拓香港界址專條), also known as the Second Convention of Peking, when Britain gained control of the New Territories for the next ninety-nine years, an article stipulated that Chinese officials remaining in the Walled City "should continue to exercise the jurisdiction except insofar as might be inconsistent with the military requirements for the defense of Hongkong [17]". Furthermore, the terms of the Convention failed to yield to Great Britain the "permanent assumption" of administrative authority over the Walled City for the "whole of the remainder of the lease without Chinese consent [17]". A diminution in diplomatic relations between imperial China and Britain at the close of the nineteenth century consequently blurred "the presumption on the part of one country that such state of affairs would continue for another 97 years [17]". Ultimately, the British colonial judicial system exercised no sovereign control over the Walled City as outlined in the conditions of the Convention. No "limitation of time" applied to the Chinese military and civil officials positioned inside the autonomy of the Walled City [17]. As late as 1959, Chinese officials asserted that they stood justified in exerting "both civil and criminal" jurisdiction over the tract of land [17].

The clause within the Convention reserving to the Chinese the rights over the Kowloon Walled City prompted outrage within the Hong Kong press and European business establishment. The Hong Kong General Chamber of Commerce opined that continued Chinese jurisdiction in the Walled City would "prolong the 'moral danger' that the City's notorious suburbs represented and 'could not fail to exercise a malign influence on the minds of the natives [15] (p. 68)'". A British journal, meanwhile, ratcheted up condemnation over the clause: "The Chinese officials must go, and the sooner they go, the better for all

concerned [15] (p. 68)". The fraught territorial status of Kowloon Walled City between China and Great Britain fundamentally transformed Hong Kong power dynamics for the next century.

The year Eighteen ninety-nine marked an overhaul in administration over the Kowloon Walled City and a shift in international relations thereafter. In April of that year, responding to the Hong Kong Government positioning itself to occupy the "New Territory", the Viceroy of the Two Guangs dispatched 600 soldiers into the Walled City [14] (p. 37). The Colonial Office under the authority of Hong Kong Governor Sir Henry Arthur Blake became irate and threatened to besiege the garrison if it remained within the City. The Chinese compensated by retaining two hundred troops inside the city as of 4 May. Despite the Chinese countermeasure, the fortification exploded into unrest on 16 May 1899. British forces, specifically three hundred men comprising the Royal Welsh Fusiliers alongside one hundred Hong Kong Volunteers, muscled into the Walled City to retaliate against the perceived duplicitousness of the Viceroy of the Two Guangs for cooperating with the grassroots insurrection. Despite protestations by T'an Chung-lin (譚鍾麟), the Viceroy, and the *Tsungli Yamen* (總理衙門), the governmental body dictating foreign policy in imperial China during the late Qing Dynasty, the British Minister ruled there would follow no Chinese reinstatement of administration over the Walled City [14] (p. 37). Chinese officials and troops consequently departed from Kowloon Walled City in humiliation, relegating the tract of land long under their oversight to a colonizer who, in turn, administered the abandoned Walled City like a derelict eyesore of colonialism.

Jackie Pullinger, an English Protestant missionary and founder of St. Stephen's Society, spent much of the later twentieth century working with at-risk individuals deep within the bowels of the infamous Kowloon Walled City. She summarized the Walled City's grey area of legality: "The Walled City did have a strange status and a peculiar life of its own: it was not governed by law. As a result, it had become a haven for illegal immigrants, criminals and vice of every kind [18] (p. 5)". Following Great Britain's unilateral seizure of the Walled City in 1899, Chinese officials on the mainland still believed the tract of land housing the former garrison was theirs. Despite staging a futile campaign of resistance that met a swift and crushing defeat, China, while disregarding the Walled City, continued to covet the land and its strategic location within the British-ruled New Territories in a diplomatic fashion. To relieve the British public's wariness of a lingering armed Chinese presence in the middle of Hong Kong Colony that harbored the potential of a skirmish similar to the one the British military forcibly put down seven months prior, the *Walled City Order in Council* of December 1899 observed and demanded:

> The exercise of jurisdiction by the Chinese officials in the City of Kowloon having been found to be inconsistent with the military requirements for the defence of Hong Kong, it is expedient that . . . the Chinese officials within the City of Kowloon should cease to exercise jurisdiction therein, and that the said City of Kowloon should become part and parcel of Her Majesty's Colony of Hong Kong [18] (p. 6).

In the years following the Second Convention of Peking and the Chinese military's ouster from its garrison post at the fortification, Kowloon Walled City fell into disrepair and desolation. The long-standing Chinese walls eventually crumbled, leaving the wasteland to be occupied mostly by incoming Protestant missionaries [14,18] (p. 38; p. 11). Outside of religion, the void left by the British became quickly consumed by illegal gambling dens decried though, nevertheless, tolerated by the British. That the British would destroy a two-hundred-year-old Chinese Kowloon military post only to leave nothing meaningful in its wake for the collective good of the Colony exacerbated Chinese indignation over British colonial administration. Britain's action also foreshadowed how future installments of Hong Kong's government would treat those regarded as public nuisances as a vehicle for reshuffling offensive sectors of the population about the Colony. The Walled City, specifically its two large cannons near the south gate that had avoided demolishment and sale following the British occupation, subsequently became a lionized Disneyfication of

itself. The site provided tourists and Hong Kong residents, alike, with a manufactured "Old China" nostalgia [14,15] (p. 39; p. 72). The Hong Kong government later evicted lingering squatters and demolished buildings in the face of Chinese opposition, however not before Canton authorities had sent "agitators" into Kowloon Walled City in 1934 to stir up the squatters to thwart resettlement efforts [10] (p. 64). The squatters continued to cite that the Walled City still officially belonged to China outside of British authority per the 1898 convention [10] (p. 64).

During World War II, under occupation by the Japanese Imperial Army, the last remaining original wall of the Walled City was leveled to extend Kai Tak International Airport (啓德機場) [19]. Nonetheless, waves of refugees poured into Hong Kong, including the tract of land comprising the former Walled City, between the cessation of Japanese occupation at the end of the war and the communist takeover of mainland China in 1949. The population of the Colony skyrocketed from 1.6 million in 1946 to 2.36 million by the end of 1950, including over 700,000 refugees who had fled mainland China in the first half of 1950 alone [10] (p. 131). Despite its internal strife, mainland China firmly clutched onto its territorial rights over the Walled City. The PRC barred the Hong Kong government from interjecting in the affairs of the once more burgeoning semi-legal grey zone in the heart of a metropolis upon the threshold of an unprecedented postwar economic boom. Pullinger notes the irony of China's hardline stance over its sovereign administration of the Walled City, noting "the ultimate expulsion of the Chinese from the Walled City, once the British had leased the surrounding 'New Territories,' paved the way for an administrative no-man's-land [18] (p. 11)". Long gone were the walls. In their place, a monolith of self-sufficient semi-legality in direct defiance to British Hong Kong had sprouted.

From its postwar rebirth until its demolishment in the early 1990s, the Kowloon Walled City mushroomed into a refuge and an opportunity at socioeconomic mobility for many migrants to Hong Kong. Kowloon accommodated a multiplicity of economic livelihoods spanning the spectrum of legality and illegality. Triad criminal operations, opium divans, and prostitution dens abounded but so did daycare centers, single room family-run businesses, unlicensed dentist offices, multigenerational families living under one roof, and a strong sense of community. The number of Kowloon's on-site squatter encampments rose exponentially in the mid-1940s. By 1971, an estimated 10,000 people occupied 2185 dwellings [19]. The fourteen-story labyrinth's poor ventilation and lack of sunlight prompted inhabitants to refer to Kowloon as "Hak Nam" or the city of darkness [15,18] (p. 118; p. 11).

Though triad gangs, including the *14K* and *Ging Yi* (京藝), controlled all organized crime inside Kowloon, the notion that police never entered Kowloon or, if so, only by force is a falsehood [15,18] (p. 272; pp. 12–13). For instance, a memorandum written for the Governor by the Commissioner of Police and also distributed to the Secretary of State for the Colonies on March 4, 1955, nullified the myth of an absence of law enforcement as the memo detailed the "extent of criminal activities" in Kowloon Walled City over the duration of the 1950s [15] (p. 272). Furthermore, "two-man foot patrols" inside the City were regular happenings with the only absence of which occurring from the late 1940s to the early 1950s [15] (p. 272). Police made arrests and intercepted ample contraband, but they proved impotent in prosecuting those arrested "for fear of incurring the displeasure of the Chinese Government [15] (p. 273)". Despite the police presence, the Triads—representing approximately 50 "known societies" and a disputed membership estimate of 100,000—embedded themselves into every level of Hongkongese society with many tentacles controlling legal and illegal businesses inside the Kowloon maze [18] (p. 15). Indeed, Kowloon represented a high-definition contrast of lightness and darkness, security and vulnerability, and thriving outside the law while looking the other way.

The Hong Kong Government in 1986 began planning for the "clearance" of the Kowloon Walled City in coordinated phases [20] (p. 258). By that time, the western and eastern parts of Kowloon City District had grown worlds apart [20] (p. 258). The "Draft Agreement between the Government of the United Kingdom of Great Britain and

Northern Ireland and the Government of the People's Republic of China on the Future of Hong Kong" signed on 26 September 1984, had recognized the Walled City's long-standing "special status" within Hong Kong's urban affairs and colonial governance [20] (pp. 263, 265). Gordon Jones, the District Officer for the Kowloon City District from November 1983 until May 1986, noted the only official population survey of the Kowloon Walled City came in January 1987 counting approximately 33,000 residents [20] (p. 274). That figure proved "substantially less than the earlier unofficial estimate of 45,000 [20] (p. 274)". Responding to the government's decision to shut down Kowloon on 14 January 1987, the Chinese Ministry in Peking (北京) released the following statement:

> Like other parts of Hong Kong, the Kowloon Walled City is a question left over from history. The governments of the People's Republic of China and the United Kingdom signed on 19 December 1984 the Joint Declaration of resuming the exercise of sovereignty over the whole area of Hong Kong by the Chinese government as from 1 July 1997. Thereby this had created conditions for fundamentally improving the living environment of the inhabitants of Kowloon Walled City. We wish to express our full understanding of the decision made by the British Hong Kong Government to take appropriate measures to clear the Kowloon Walled City and build it into a park [18] (pp. 107–108).

Gutting Kowloon Walled City affected the financial livelihoods of everyone inside. Those who lost their businesses included unregistered doctors and dentists as well as plastic, garment, and food-processing operations often run by families in a single room. The government considered these lost livelihoods when pondering the monumental task of clearance and rehousing [18] (p. 109). Under the oversight of the Hong Kong Housing Authority, the government compensated the 33,000 residents and businesses with HKD 2.7 billion (350 million USD) [20] (p. 275). The 18 January 1987, issue of the *South China Morning Post* ruminated the most suitable memorialization of the Kowloon Walled City:

> Before the wreckers obliterate everything, some effort should be made to peg the boundaries of the original wall which for centuries protected the residents of the area from wandering brigands and pirates. *Why not rebuild the wall?* . . . [In preserving the old cannon] They should be preserved, if only as a monument to the independent spirit that has flourished among the people there, an independence that was illustrated some years ago when residents there foiled attempts to move the cannon . . . The dentists and their displays can go. But there is no need to wipe out forever all traces of what once was a bit of Hong Kong's history—an important bit [18] (pp. 112–113).

By March 1993, the entire Walled City—comprising just 2.8 hectares containing 350 sardine-packed buildings occupied by 60,000 dwellers at its apex—stood empty [18,19] (p. 124). Demolition began on 23 March 1993, and concluded in April 1994 [20] (p. 275). Amid the void, a few relics had survived. The former *yamen* (衙門) building of Kowloon Walled City and the vestiges of the South Gate were memorialized on 4 October 1996, with the Kowloon Walled City Park dedicated on 22 December 1996, by the final Governor of Hong Kong, Chris Patten [20] (p. 275).

The legacy and memory of Kowloon Walled City is as grey as its former legal status. After driving out the Chinese, the unsystematic administration of the Walled City bruised British paternalism. Afterward, the Walled City symbolized a Cold War-style standstill between the begrudging British and a steadfast China unwilling to relinquish control of the small territory. Many found relief that the sordid symbol of lawlessness was finally gone. Still, others fondly remembered Kowloon for its tight-knit community, entrepreneurial temper of its inhabitants, and as a refuge for scores fleeing Communist China. Kowloon Walled City typified a unique labyrinth subverting the notions of sovereignty and international relations. More importantly, it personified how a structure, itself, can serve as both a window into the troubled relationship between the Hong Kong government and its marginalized denizens as well as a guidepost charting the path of Hong Kong's overall de-

velopment. Despite its demise, similar living arrangements and self-agency once embodied by the Kowloon Walled City continue in modern-day manifestations worldwide.

## 4. Chungking Mansions

Anthropologist Gordon Mathews, who spent 2006 to 2010 analyzing and absorbing the communal patchwork of human relations and commerce teeming inside Chungking Mansions, called the notorious structure—a "cramped area of 100 meters square" comprising "90 guesthouses and 380 businesses in the building"—a confluence of "low-end globalization [5,6] (pp. 19–20; pp. 169–170)". At a height of seventeen stories, located on the "Golden Mile of Nathan Road" in Tsim Sha Tsui, Chungking sees approximately 10,000 people representing over 100 nations streaming inside of it daily [5,6] (p. 7; p. 170). Mathews defines "low-end globalization" as the "transnational flow of people and goods involving relatively small amounts of capital and informal, sometimes quasi-legal or illegal transactions, commonly associated with the developing world [6] (p. 170)".

One may obtain everything in Chungking Mansions from lodging to a haircut, whiskey to sex, computer repairs to electronic gadgets, groceries to legal advice for asylum seekers, and "spiritual sustenance" for Christians and Muslims [5] (p. 27). Chungking Mansions, containing hundreds of businesses and vendors, showcases four entrepreneurial types: traders, owners/managers and their employees, asylum seekers, and tourists [6] (p. 171). Diverse viewpoints of Chungking acknowledge its spiritual energy and "ethnic chaos [6] (p. 175)". Additionally, dozens of sex workers (mostly Chinese and Indian) ply their trade within the Mansions on any given day as do heroin addicts and drug dealers who often sleep in the neighboring alleys [6] (p. 171). Nevertheless, all these groups function in relative spatial and temporal cohesion. Hong Kong's laissez-faire capitalism, easily obtainable tourist visas, deregulation, and individualism have all combined to engineer a "global peace" in Chungking Mansions where the absence of violence is attributed to "the common pursuit of profit by all who sojourn there [6] (p. 178)".

Chungking Mansions first opened in 1962 [5] (p. 3). As Mathews comments, Chungking ostensibly stands as a "dilapidated seventeen-story structure full of cheap guesthouses and cut-rate businesses in the midst of Hong Kong's tourist district [5] (p. 7)". More importantly, the Mansions offers a sense of "adventure" and unbridled capitalist initiative for entrepreneurs and temporary laborers, notably from South Asia and sub-Saharan Africa, alongside asylum seekers and tourists seeking cheap lodging in the epicenter of Hong Kong's commercial glitz [5] (p. 7). Approximately 4000 people stay in Chungking Mansions per night [5] (p. 7). Opposite the ultra-sleek skyscrapers of Hong Kong via posh Nathan Road, Chungking Mansions appears "terrifying" to many Hong Kongese [5] (p. 13). Members of Chinese-language blogs and chat rooms have noted trepidation when walking past the structure, thinking they could lose their way inside the building and risk kidnapping. One internet user commented: 'I am ... afraid to go. There seem to be many perverts and bad elements there [5] (p. 13).' Most of Chungking Mansions' global critics on the Internet, comprising majority American and European viewpoints, deplore the structure as the "sum of all fears for parents whose children go backpacking around Asia" while denouncing it as "a den of crime, of drug trafficking, prostitution and generally all the nastiness that goes on in the world you can find in Chungking Mansions [5] (p. 14)". Xenophobic viewpoints expressed by the Hong Kong Chinese and others from the developed world, apprehensive over the throng of people from the developing world congregated in and around the structure, have primarily contributed to Chungking Mansions' symbolization as an unsavory, dangerous, and exotic locale. While being a "historically incongruent" comparison to Kowloon Walled City regarding its "off-limits to external authorities", the contemporary Chungking Mansions situates itself "*in* Hong Kong, but ... not *of* Hong Kong. It is an alien island of the developing world lying in Hong Kong's heart [5] (p. 15)".

Chungking Mansions became a magnet of attention in Tsim Sha Tsui from the moment it opened its doors. A Hong Kong real estate intellectual once remarked: "It was high-class in the 1960s because it was so tall—that area didn't have tall buildings then, and Chungking

Mansions definitely stood out [5] (p. 33)". Nevertheless, notoriety was not far behind as others disputed Chungking Mansions' "high-class origins [5] (p. 34)". A British man, who resided there from 1962 to 1964, said: "It didn't seem high-class at all . . . It had a seedy atmosphere. It was built cheap [5] (p. 34)". Chungking's feeble design and labyrinth interior conjured the impression of a "rabbit warren". Chungking follows the conventional "vernacular design" commonly found in Hong Kong in the 1960s involving a "two-level plinth and tower blocks rising above it [5] (p. 23)". "[Three] blocks and five separate elevators" comprise Chungking Mansions' architecture, each connected to "their own separate worlds" and inaccessible except via the ground level [5] (p. 23). Approximately 140 retail outlets at the base level offer a "dizzying" array of merchandise [5] (p. 25). Unlike ethnographies examining groups of people and despite its loathsome presence, Chungking Mansions is in itself a symbol and convergence point for uninhibited globalization.

Gordon Mathews' 2011 publication, *Ghetto at the Center of the World*, first documented the history of Chungking Mansions. Three prime reasons account for the existence of Chungking: the affordability of business and residence, the "ease of entry" into Hong Kong for overseas entrepreneurs to purchase goods wholesale, and the proximity to southern China as a global manufacturing hub [5] (p. 16). Visitors to Hong Kong enjoy "comparatively relaxed" visa procedures compared to those entering mainland China with visa-free entry into Hong Kong for upwards of ninety days [5] (p. 17). Entrepreneurs across the developing world converge upon Chungking Mansions to purchase Chinese goods and then often continue to mainland China to buy additional goods wholesale from the source to transfer back to Hong Kong before finally returning to their home countries. These merchants continue to flock to Chungking Mansions because it embodies what Mathews calls a "very concentrated picture of low-end globalization in a very small place [5] (pp. 18, 21)".

The area of Tsim Sha Tsui, a district on the southern point of Kowloon Peninsula teeming with shopping and nightlife, has long been an epicenter of multiculturalism and a hotbed of rough-and-tumble transients. A noticeable South Asian presence appeared in Tsim Sha Tsui from the mid-nineteenth century onwards with a sizeable Indian demographic representing the Hong Kong Crown government as policemen and soldiers. In the early-to-mid 1960s, nightclubs, a shopping mall, and jewelry shops dominated the business establishment of Chungking frequented by Hong Kong celebrities and British Army officers, many of whom resided in the building [5] (pp. 33–34). American service members rotating out of the Vietnam War in the late 1960s contributed to the transformation of Tsim Sha Tsui from seemingly high-class into a well-known red-light district [5] (p. 34). *Lonely Planet* guides and Tony Wheeler's *Southeast Asia on a Shoestring* brought the image of Chungking Mansions to a broad Western audience. Moreover, the 1981 edition of Wheeler's guidebook boasted "there's a magic word for cheap accommodations in Hong Kong—*Chungking Mansions* [5] (p. 35)". The 1970s saw Chungking continue as a fashionable locale for Western bohemians, beatniks, and backpackers to congregate. From the 1980s until the early 1990s, the seedy ambiance exhibited by Chungking and its colorful denizens grew even more substantial with its portrayal in Chinese film director Wong Kar-wai's (王家衛) 1994 Hong Kong comedy-drama film, *Chungking Express* [5] (p. 14).

Fires were a constant at Chungking with one of its most notable occurring in February 1988 on the 11th floor, killing a Danish traveler as he tried to escape by diving out of the window on a mattress [5] (p. 36). Subsequent worldwide headlines—including "High-Rise Menace Needs Urgent Action by Govt", "Chungking 'to Remain a Firetrap'", and "Facelift Fails to Improve Fire Safety at Chungking"—clamored for stricter safety standards [5] (p. 36). In 1993, Chungking's image was further sullied when its power system overloaded and exploded, causing a blackout [5] (p. 36).

By the turn of the twenty-first century, Chungking Mansions had accrued a less nefarious image much to the lamentation of some travelers who expected an unconstrained atmosphere. One French backpacker wistfully commented: "It used to be so crazy . . . when I was here in the early 1990s. Additionally, now it seems so normal, so bourgeois [5]

(p. 196)". In the 1980s and 1990s, Pakistani gangs filled the Triad absence [5] (p. 36). However, throughout the 1990s, police raids extinguished much of the organized crime element in and around Chungking while also evicting many Bangladeshis, Indians, Pakistanis, and Nigerians for lacking proper documentation. Concurrently, a massive influx of Africans ventured into Chungking with the compulsion to buy Chinese goods in Hong Kong and China before selling them at a profit in their home countries [5] (p. 37). Within reach, Chungking Mansions still epitomizes the 'exotic third world in a safe first-world city [5] (p. 195).

Chungking Mansions symbolizes vivacious globalization, and its inhabitants encounter a slew of new experiences there. Those flocking to Chungking for money, adventure, or a bed continue to enter the imposing structure because Hong Kong and China convey an aura coaxing them to seek fortune and memories. Mathews elaborates on the diversity of Chungking's inhabitants and its enduring entrepreneurial zeal:

> Not just a third-world enclave but a middle-and upper-class third-world enclave of people with money and education far beyond that of most of their fellow citizens at home ... the fact that almost everyone in Chungking Mansions is a comparative success in life, by the very fact that they are in the building. Most are successes ... within the social-Darwinistic competition of neoliberalism ... It is ironic that a building popularly viewed in Hong Kong as a cesspool of sleaze is in fact a staunch bourgeois enclave of chamber-of-commerce capitalism, albeit with a few corners cut [5] (p. 213).

Chungking Mansions stands as a lasting tribute and prioritization of "face-to-face relations" among its buyers and sellers [5] (p. 209). A Hong Kong government official in 1993 surmised Chungking Mansions would never undergo redevelopment because "even in its current condition, the building is a gold mine for its owners [5] (p. 217)". Despite being an engine of socioeconomic mobility, experts on Chungking Mansions, notably Mathews, as well as Hong Kong urban studies scholars, believe the structure will inevitably be replaced by glitzier accommodations more in keeping with the architecture lining Nathan Road. Regardless of its future, Chungking Mansions represents global migration, entrepreneurship, free-market economics, and international relations between the developed and developing worlds from the ground up. On a local scale with global implications, the everyday transactions and interactions of its inhabitants redefine and reinvent twenty-first century concepts of borders, public spaces, subsistence, and the outward image of Hong Kong.

## 5. Squats, Tenements, and Fires

Associated with overcrowding, both squatter encampment and tenement fires posed persistent sociocultural predicaments in Hong Kong spanning the nineteenth and twentieth century. Squatters always drew condemnation and paternalist concern from the press. In addition, colonial officials routinely sought strategies to negotiate the squatter predicament predominately involving expulsion, demolishment, and relocation. The British and their Chinese counterparts in Hong Kong's administrative apparatus recognized that a fundamental component of their civilizing mission in the Colony rested upon the accessibility of descent housing. However, officials prioritized neither an adequate supply of housing, land on which to build, nor efficient construction. From 1945 to 1954, "no more than 200 acres of crown land were allocated to new urban housing, and less than thirty acres of this were auctioned on the open market [9] (p. 14)".

In November 1946, *The China Mail* published an article titled "Hong Kong Squatter Problem", spotlighting commercial and labor representatives at the Chinese Chamber of Commerce discussing squatter and unemployment issues [21]. The meeting, requested by the Government in response to the rising number of unemployed squatters living on the streets and in "partly demolished houses", resolved the "repatriation" of squatters and others without employment or financial support was deemed a "necessity [21]". Just three Chinese representatives were present at the meeting. At the time, Hong Kong's

estimated 20,000 vagrants, considered a "grave menace created to public health", represented a "community problem" for which one Chinese community requested assistance in crafting a "solution [21]". If the squatters proved unable to find work in the Colony, the government stood ready to dole out monetary assistance if they repatriated back to the Chinese mainland. However, for the operation to be effective, conditions in China would have to demonstrate noticeable stability and economic prosperity to entice those cajoled by the Hong Kong government to vacate the Colony. Meanwhile, others opposed repatriation efforts and demanded the government build affordable housing "or huts" in Hong Kong's New Territories to shelter the destitute and unemployed [21]. The motion passed despite many abstentions. It pushed for "Chinese organisations themselves to urge their less fortunate compatriots to return to their native villages [21]".

"Startling revelations of poverty and overcrowding are contained in the fifth annual report of the Society for the Protection of Children" began an article in *The Hong Kong Telegraph* in late 1934 detailing the rampant poverty, disease, and food shortages found in the Colony's tenements [22]. The Society was principally involved in the redressing and promotion of children's welfare, having worked thousands of cases over the previous five years. In Hong Kong's Western District where the earliest tenements teemed, higher population density as compared with Kowloon and Eastern Hong Kong produced the "least hygienic" conditions [22]. The "average monthly income per head" in cases examined by the Society factored out to HKD 2.93 in 1932 [22]. Across Hong Kong from Kowloon to Western and Eastern Hong Kong, average monthly income per household shrank from 1931 to 1934. The Society's report additionally noted the excessive overcrowding of tenement houses, detailing a case involving a "ground floor level where six families were accommodated, while eight women slept along the walls of the 'bed spaces' and four more had the comparative privacy and comfort of the 'cock loft [22]'". A cockloft is a small loft or attic situated above the highest finished ceiling of a building. Between 1921, the year that the Hong Kong Government imposed rent controls, and 1931, Hong Kong's population ballooned from 625,000 to 850,000 [23] (pp. 27, 33). The Society also observed instances of cramped living arrangements revealing "22 adults and eight children occupying a floor space of about 700 square feet, sharing the same kitchen, with no sanitary arrangements whatever in the building [22]". The "average rent of a bed-space" stood at HKD 2.71 in Hong Kong and HKD 1.92 in Kowloon with the "average rent of cubicle" in Hong Kong being HKD 3.64 compared to HKD 3.19 in Kowloon [22]. The Society's report consequently elicited momentum to demolish the tenements with its "thorough though unobtrusive" reporting of the Colony's most impoverished neighborhoods [22]. The report envisaged the Society's civilizing mission as "a new attitude towards dirt and disease which, when slum clearance begins, will be of immense value" to all involved [22] (p. 7).

In the summer of 1948, a landmark Urban Council report submitted to Hong Kong Colonial Secretary David MacDougall represented "the first defining post-war statement on squatters and established the Hong Kong government's initial policy towards their clearance and resettlement [23] (p. 166)". By 1949, Crown government officials came to view the Colony's estimated 300,000 squatters as a "public liability as dangerous as they [were] uneconomic [24,25] (p. 33)". The perceived danger that squatters posed magnified once squatter settlements began spreading onto private property in residential areas. The squatters, villainized for their settlements known as "black [spots]", were considered a "menace of contagious diseases [25]". Three main factors drove the proliferation of squatter settlements during the first decade following World War II: the absence of low-cost private housing, high demand for multifarious living situations, and the risk taken by both squatters and construction companies to build without permission [24] (p. 33). Mindful of the control held by the Hong Kong Building Authority and Health Department, *The Hong Kong Telegraph* remarked: "It has always been our understanding, however, that no building for habitable purposes can be constructed on leased land unless it conforms to the regulations of the Building Authority and the Health Department" with which the squatter settlements did not agree [25]. Within the settlements, sanitation facilities were

almost nonexistent, ventilation was insufficient, and fire code violations abounded. All this made for "a mockery of the minimum requirements [25]". The *Hong Kong Telegraph* pushed for a revised "official review" process alongside a "new policy" in addressing the plight of the squatters [25]. "To allow unrestricted squatting on Crown land or leased sites", an editorial in the newspaper read, "is but taking the least line of resistance and serves only to consolidate the problem of squatters and public health [25]". Proportional distribution of the squatter "reservations" to certain areas of Hong Kong Island, Kowloon, and the New Territories would rationalize the "evil" while concurrently allowing for a "degree of supervision [25]". Nevertheless, shuffling squatters around the Colony still proved a "losing game [25]". Placing squatters within a controlled, regulated area with defined borders would render the squatters as "less of a public menace [25]". Once again, *The Hong Kong Telegraph*'s call for the eradication of squatter settlements, while linking the squatters' substandard living conditions to "the presence of unsanitary huts in residential areas", goaded government officials to orchestrate "their immediate removal" for the benefit of the Colony [25].

The 6000 "refugees" at Rennie's Mill, like all Hong Kong squatters, were viewed as a drain on the Hong Kong taxpayer to the tune of HKD 180,000 per month [26]. The Crown government, the press noted, "would welcome their departure" when considering maintenance costs and "its inmates [26]". "The Government would be grateful to be rid of people neither useful to themselves nor to the community, and any different attitude would rank as stupid, but deportation has never been contemplated", the editorial in *The Hong Kong Telegraph* chided [26]. The newspaper alleged the "magnitude" of the squatter dilemma involved approximately "half a million [26]". The editorial postulated that less than one tenth of the 500,000 could be considered "normal residents" holding any "real claims on the Colony [26]". The solution rested on a "radical" shift in Hong Kong politics to initiate the utilization of vacant Crown lands to serve the "mutual benefit" of the entire Colony [26]. An article similar in tone, appearing in the "Comment of the Day" section of *The China Mail* in February 1953, opined that if the committee and the Urban Council could "speed up fulfillment of the squatter resettlement programme, it will be doing a great service to the community [27]". "The whole squatter problem is one of great urgency calling for quick action", the newspaper proclaimed, "and the hope now is that with the whole of the Urban Council devoting its attention to the problem, it will be all the more readily resolved [27]".

Throughout the 1950s, a slew of intense fires rampaged through Hong Kong's squatter settlements, killing few but leaving tens of thousands homeless. A notable conflagration on 21 November 1951, scorched squats in Tung Tau (東頭) to the northeast of Kowloon Walled City [9] (p. 73). The following day, the *South China Morning Post* "suggested that over 3000 huts housing about 15,000 people were destroyed", while the *Sing Tao Daily* (星島日報) estimated the fire consumed approximately 20,000 homes [9] (p. 74). On 12 January 1950, an inferno in the Kowloon City district thought to be the largest in the Colony's recorded history burned for over four hours across nearly one square mile containing three squatter villages [28]. Between 1500 and 2000 huts were incinerated, leaving approximately 10,000 people destitute [28]. The Hong Kong Fire Brigade, the Army's fire engines, and Kai Tak Airport's RAF station all joined in fighting a blaze exacerbated by a lack of fire hydrants and a sprawl of tightly packed huts [28]. In the thick of the blaze, when incinerated huts storing flammable materials detonated, looters were reported as "preying on the helpless victims . . . mostly women and children [29]". Fires such as the January 1950 blaze, while tragic and costly, also fortuitously provided a powerful impetus for government officials and the press to finally tackle the squatter population and tenements at full force. These developments notwithstanding, a catastrophic fire nearly four years later earnestly initiated the groundswell of government intervention regarding the Colony's squatter dilemma. All told, at least 190,047 people lost their homes to squatter fires during the 1950s according to official estimates [9] (p. 2).

After rendering more than 50,000 tenement dwellers homeless in one night, the Shek Kip Mei (石硤尾) fire of 25 December 1953, prompted two urban planning responses: unregulated high-density housing and land advocated to "be less than that occupied by the same number of people in squatter conditions [9,30] (p. 95; p. 610)". Astoundingly, only two people died in the Shek Kip Mei fire [9] (p. 95). Both the PRC and the United States notably sent aid to the victims. While the blaze finally spurred idle Hong Kong officials into action, anthropologist Alan Smart differs from the notion that the Shek Kip Mei fire fundamentally spurred an instantaneous "momentous decision" in shaping Hong Kong's public housing system [9] (p. 114). Smart posits "other, mostly forgotten, squatter fires, both before and after Shek Kip Mei played crucial but neglected roles in shaping Hong Kong housing policies, and hence the very nature of contemporary Hong Kong itself [9] (p. 3)". Far from being the last settlement to succumb to flames, Smart postulates the Lei Cheng Uk (李鄭屋) fire was the final conflagration of the 1950s to impact Hong Kong's housing and resettlement policy significantly [9] (p. 157).

Manuel Castells, a sociologist and urban studies scholar, contends the Shek Kip Mei fire heralded a full-fledged public housing system in the Colony [31] (p. 1). From 1955 to 1968, the Hong Kong Resettlement Department erected 466 "resettlement blocks" as a response to the Shek Kip Mei fire [30] (p. 610). The fire was a landmark event that galvanized public policy, embodying "social paternalism of the administrative class [31] (p. 150)". Hong Kong's disadvantaged demographic would receive public assistance, municipal budgetary limitations would receive adherence, a bureaucracy would operate efficiently, and economic growth would take precedence [31] (p. 150). Without question, the Shek Kip Mei fire and consequential government decisions in the 1950s proved paramount in incentivizing the present composition of Hong Kong's public housing system. Government reaction to the fire confirmed the far-reaching influence of the Hong Kong Housing Authority and elevated the governmental body to dictate the living arrangements for millions of people.

## 6. Hong Kong Public Housing and the Hong Kong Housing Authority

Hong Kong's postwar economic recovery translated into a "miracle" of modern infrastructure across the Colony [10] (p. 132). Leo F. Goodstadt, a Hong Kong-based British economist and the first chief of Hong Kong's Central Policy Unit from 1989 to 1997, points that Hong Kong's "intractable housing woes" began in the 1940s with waves of new arrivals establishing squatter huts on hillsides, rooftops, and other empty plots of land [3] (p. 93). These domiciles, however, as Goodstadt argues, "were not the result of poverty" like other squatter enclaves in the world's metropolises as Hong Kong, including its squatter demographic, flourished due to Hong Kong's industrial boom in the 1950s marked by high employment and increasing wages felt in the proceeding decades [3] (p. 93).

From 1954 until 1984, the government-induced Hong Kong Housing Authority engaged in a "resettlement" program for its dispossessed squatter and tenement dweller demographics [24] (p. 30). After the 1984 Sino-British Joint Declaration, the Hong Kong government exerted greater control over the territory's "illegal migration and irregular settlements" under a phase of urban governance that Alan Smart labels as "exclusion [24] (pp. 30, 38)". The Sino-British Agreement, in effect until 1997, lessened access to and affordability of private housing as a consequence of radical squatter protests challenging Hong Kong's housing policies in the early 1980s [24] (pp. 36, 38). Moreover, the Housing Society, "a private charitable institution", was established in 1951 [31] (p. 20). Despite the formation of these two institutions, immigration coupled with private redevelopment displaced "tenement dwellers" and worsened Hong Kong's housing crisis in 1964 as compared to 1954 [31] (p. 21). The number of squatters in Hong Kong rose from 300,000 in 1953 to 550,000 in 1964, comprising an estimated twenty percent of the Colony's population [31] (p. 21). At the same time, however, the Housing Authority became a crucial landlord in the commercial and industrial land development game [31] (p. 33). In 1964, the government

created the "White Paper" outlining an aggressive agenda to resettle 1.9 million people over the next ten years at a cost of HKD 1.7 billion attached to an additional initiative to house at "low cost" another 290,000 residents to the tune of HKD 353 million [32] (p. 11). The Housing Society reacted to the 1964 annual report commenting: "It is vitally important not to concentrate completely on minimum standard accommodation. Deplorable housing conditions are the most serious social problem, and the greatest fear menacing the people of Hong Kong is of being forced into the street with nowhere to live [32] (p. 11)".

By the mid-1960s, the influx of squatters into the Colony had temporarily leveled off due to an unprecedented resettlement program and a sharp decline in real estate activity [30] (p. 609). This development was additionally complemented by the government recording forty-three percent of the population living in government or government-subsidized housing [32] (p. 17). In 1969, the government noted that it had granted "squatter resettlement accommodation" for nearly 1.1 million people by 1968 [30] (p. 608). However, these figures lose their luster when considering Goodstadt's argument that while the Hong Kong government cleared squatter areas to allow for land development, the "actual living standards" the government provided "were still barely adequate [3] (p. 94)".

By 1970, Hong Kong counted four million people with eighty percent of its total population found in Victoria and Kowloon, exacerbating already high population densities in those areas. These overwhelming population densities urged landlords to subdivide floors in tenement buildings. These units first became "cubicles" before downgrading to "bedspaces [3] (p. 94)". Concurrently, Hong Kong's income disparity also widened from the 1950s to the 1970s with the average Hongkonger during this period working more than ten hours per day seven days per week [10] (p. 137). On the heels of the 1966 Kowloon disturbances, observers assessed:

> It is obvious that the pressures due to overcrowding in the twin cities of Victoria and Kowloon, combined with the hard struggle for a living, which is increased by the need for many of the population to assist families in China in addition to their own dependents in Hong Kong, the lack of homogeneity in the population and the underlying insecurity of life in the Colony, resulting from international political and economic conditions, create tensions which elsewhere would be more than sufficient cause for frequent disturbances [10] (pp. 137–138).

By the end of 1970, over one million people lived in "low-cost housing estates" and approximately fifty percent of Hong Kong's total population resided in "government or government-subsidized dwellings [10] (p. 132)". In 1972, Hong Kong Governor MacLehose announced a ten-year housing program aimed at providing "public housing estates" for a further 1.8 million people [2] (pp. 11–12). At the time of MacLehose's proposal, an estimated 350,000 Hongkongers still resided in squatter settlements [33] (p. 494). Additionally, in 1973, MacLehose reformed the Hong Kong Housing Authority to realize his housing development initiative [33] (pp. 494–495). Still, the postwar manufacturing boom catapulting Hong Kong to first-world status did not shield Hongkongers from steadily rising rents. The non-profit Housing Society, unaffiliated with the local government, began to raise rents every two years beginning in 1976 "when rents of over half the flats were increased by about 20 per cent [32] (p. 87)". MacLehose's housing development policy, however, managed to provide housing for only 960,000 people by 1982 when MacLehose's tenure as Hong Kong Governor ceased, leaving 750,000 Hongkongers still in squatter settlements [33] (p. 495). Still, as Hong Kong social policy scholar Andrew Yu promotes, MacLehose's ten-year housing plan proved "the most extensive housing construction plan in the history of Hong Kong", benefitting many and also spurring the Hong Kong colonial government to continue to construct public housing estates and "new towns" following MacLehose's departure [33] (p. 495).

In the early 1990s, the same Housing Society voiced concern over a "shortage of land supply" with the added hope the Government "would release more land so that the Housing Society might be able to accelerate the provision of affordable housing [32] (p. 89)". Escalating rents outpaced the Hong Kong government's willingness to allocate more land

necessary for organizations such as the Housing Society to provide more housing. While the total income for 1997, the year of the Handover, amounted to "$1.09 billion from sales and rents, the highest recorded in the history of the Society", land remained absent of urban development [32] (p. 90). As a result, housing became more unattainable for many. Out of this discrepancy arose a new living situation that crammed as many people as possible into the smallest of spaces: cage homes.

**7. Cage Homes**

Anthropologist Christopher Cheng writes that two kinds of Hongkongers resided in a *tong lau* (塘樓), an architectural style emblematic of Hong Kong constructed from the late nineteenth century until the 1960s. A *tong lau* served both residential and commercial functions. Its two kinds of residents included its owner-residents representing multigenerational migrants who had purchased the properties as well as subtenants squeezed in cage-homes and cocklofts [34] (p. 15). The latter group became the most marginalized in all of Hong Kong, subsisting behind closed doors under the disdain of the government. Mike Davis notes Hong Kong features a quarter of a million people who "live in illegal additions on rooftops or filled-in airwells in the center of buildings", of which the landlords are aware [8] (p. 35). While certainly not limited to men, dwellers of cage homes, colloquially known as "caged men", occupy "bedspaces for singles, the 'cage' suggested by the tendency of these tenants to erect wire covering their bed spaces to prevent theft of their belongings. The average number of residents in one of these bedspace apartments is 38.3 and the average per capita living space is 19.4 square feet [8,35] (p. 35; p. 63)". Housing studies experts John Doling and Richard Ronald define "'cage-man' apartments [as] triple-deck bunk beds enclosed like a cage [36] (p. 79)". These "'subdivided' flats are converted flats in tenement blocks in inner city areas", containing no formal approval process and minimal health and fire safety compliance [36] (p. 79).

In July 1998, the government initiated the "Bed Space Apartment Ordinance", principally stipulating that residents' noncompliance to abide by fire safety standards and requirements would result in homelessness [37]. A report in the *South China Morning Post* in May of that year titled "An End to the Shame of Cage Homes" proclaimed: "for years . . . slumlords netted profits from packing human beings into tenements sub-divided into tiny coffin-like squares [37]". In some instances, "squalid" cage homes were also incorporated into guided city tours [37]. However, the government's Home Affairs Department, along with the Land Development Corporation and Architectural Office staff, sought change when it proposed a "purpose-built 16-storey structure" containing "312 well-designed individual homes for caged men" at the cost of HKD 60 million [37]. Ms. Lau, a representative of Home Affairs, noted that officials purposed the building for primarily housing "singletons": "unmarried factory workers, unemployed and unskilled men in their 40s and 50s" who lived outside the net of "social, community or housing" welfare [37]. The standard "50 square foot" domicile would contain "a bed (built above storage space) fitted cupboards and wardrobe, a desk and shelves" while additionally featuring "a fan . . . ample electricity plugs and provision . . . for air conditioners [37]". Rents would range "from $800 to $1200 for upper floors [37]".

Regarding the separation of the sexes in the new structure, Ms. Lau opined: "People may say I am sexist, because we have no plans for similar buildings for women . . . The fact is, women in this age bracket seem much more capable of looking after themselves. They can find homes to share. For men, it is much harder to cope [37]". "I don't want to run all the bed space operators out of business; some of them provide reasonable economy accommodation. But the guilty ones; well, good riddance. They are our shame", Ms. Lau concluded [37]. While civic organizations, including the Home Affairs Department, modestly transitioned Hong Kong's cage dweller demographic to dignified housing, the results remain insufficient in present-day Hong Kong.

Citing the Census and Statistics Department, Hong Kong media reported as of 2016 over 200,000 people in the SAR dwelled in approximately 88,000 "[coffin] cubicles, caged

homes and subdivisions [38]". These statistics, meanwhile, did not include "those living illegally in industrial buildings" estimated at 10,000 people [38]. Tony, one of twenty-two residents squeezed inside a five hundred square foot apartment, expressed to the *South China Morning Post*: "The most difficult thing about living here, is not being able to breathe in fresh air. It's suffocating [38]". Reminiscent of housing and government officials attempting to remove squatter settlements in the early twentieth century, the Hong Kong government in 2016 initiated a "plan to [criminalize] landlords operating illegal cubicle flats in industrial buildings [39]". The "plan" carried the potential to leave thousands homeless "if authorities [failed] to lay down a comprehensive resettlement plan [39]". The government had attempted a similar measure once before in the 1990s when "thousands" of cage home inhabitants were ousted from their residences owing to "new licensing legislation [39]". Those subsisting "illegally in the city's 1900 industrial buildings" likened the arrangement to a "refugee camp—cramped, humid and with poor ventilation [39]". Many Hongkongers such as Tony floundering in an average waiting time of four years for public housing, dizzying property values, and swelling rents in Hong Kong's private sector desire access to two universal aspirations: a job and public housing [38,39].

In 2019, amid the turmoil embroiling the SAR initially over the mainland extradition bill, Hong Kong featured "the world's most expensive housing market [40]". According to journalist Yonden Lhatoo of the *South China Morning Post*, as of 2019, the SAR boasted property values that had risen by 126 percent since the 1997 Asian Financial Crisis [40]. Lhatoo vented that mortgages had become "bottomless pits" consuming approximately "70 per cent of incomes among those who are desperate or foolhardy enough to buy homes in these ridiculous times [40]". The "antiquated system", Lhatoo argued, predicated itself upon the government's overreliance on land sales resulting in "[hundreds] of thousands [packed] like sardines into tiny subdivided flats because they can't afford the high-rise pigeon holes that the more privileged among us are cooped up in, or qualify for public housing which has waiting lists long enough to ensure people die in the endless queues [40]". "The solution", Lhatoo concludes, "is a political one and any revamp of the system will require the courage to take on powerful property tycoons who continue to game it, whether by hoarding land or putting 'homes' less than 200 [square feet] in size on the market, passing them off as fit for humans [40]". In a cosmopolitan city revered for its progress and rugged individuality teetering between foregone British colonialism and present-day Chinese dominion, Hong Kong remains incredibly divided over an array of long-standing issues. Housing, chief among the grievances, is a baneful crux to all other social, political, and economic ills afflicting the SAR. Hong Kong Chief Executives since the 1997 Handover maintain a policy of "minimum government intervention in the housing market regardless of the social and economic costs" evocative of unending overcrowding and squalor within one of the world's most advanced economies [3] (p. 103).

## 8. Conclusions

This paper attempted to argue that the three cases of 'heterotopias' examined above—Kowloon Walled City, Chungking Mansions, and cage homes—historically and contemporarily tether Hong Kong to its public housing system and housing woes. While perhaps murky, underappreciated, or ignored in Hong Kong's public memory, the three living arrangements, as well as the rash of conflagrations that scorched squatter settlements, have all shaped Hong Kong to be the prominent megacity it is today. Does a vestige of Hong Kong's colonial past such as the Kowloon Walled City still hold relevancy to the metropolis's present composition? May the legacy of the SAR's public housing system be viewed as a successful outgrowth of the socially maligned tenements and slums of yester years? Are modern-day cage homes a reminder of a not-too-distant past or a hint into Hong Kong's future? Or both? These three precarious living situations, integral in the trajectory of Hong Kong, illustrate circumstances when city ordinances and laws are flouted, when people cram into tiny spaces at extraordinary levels, and when the fabric of the city tolerates the existence of these residential arrangements until a minority governmental consensus

disaffects a noteworthy population sector. While these structures conjure images of vice and socioeconomic disenfranchisement, they also represent a tenacious drive to succeed through arduous circumstances while keeping family and independence in perspective. They also evidence fault lines between the social classes, British and Chinese civilization, and the Hong Kong government pitted against its general populace. This paper intended to capture what these structures, themselves, represent both to the progression of Hong Kong, itself, and the image of Hong Kong they contribute to the world.

Hong Kong showcases a highly developed economy, often bridging the choppy geopolitical gulf between the world's two largest economies: Communist China and the United States. With remarkable economic evolution comes widening income inequality as the twenty-first century progresses, and housing serves as a barometer of Hong Kong's socioeconomic polarization as well as a portal into its colonial past. Upon the 1997 Handover between the two nations, China promised Great Britain that it would respect Hong Kong's agency through the "one country, two systems" agreement until 2047 when the fifty-year period observing this constitutional principle officially ceases. Indicative of the "one country, two systems" model is the Basic Law of Hong Kong once referred to as "the blueprint for both [the] survival and well-being" on Hong Kong's political, economic, and societal fronts [3] (p. 195). The proposed extradition bill of 2019, as with the 2003 National Security Bill and a 2014 PRC State Council white paper titled "The Practice of the 'One Country, Two Systems' Policy in the Hong Kong Special Administrative Region", exposed the PRC hastening 2047 with imperiousness [41] (pp. 467–468).

Beijing clamped down tighter in mid-2020 when the Standing Committee on the National People's Congress (NPCSC) unanimously passed a new Hong Kong security law broadly criminalizing acts of secession, subversion, terrorism, and foreign collusion in the SAR. Equally as concerning as the contents of the new security law is the fact that neither Hong Kong officials nor the public knew of the new law's contents before its passing. Hong Kong's citizens are right to be mindful of a clear, full-court press from mainland China into the SAR's affairs. Hong Kong's many societal freedoms, privileges, and distinctions will never be the same again. Notwithstanding, the dilemma of public housing and unorthodox dwellings in Hong Kong, regardless of Beijing's intrusion into Hong Kong's civil liberties, will continue to sow division within the SAR. Hongkongers ought to realize that many of their societal ills, notably housing inequality, remain chiefly up to them to resolve despite political contestations with the mainland.

**Funding:** This research received no external funding.

**Institutional Review Board Statement:** Not applicable.

**Informed Consent Statement:** Not applicable.

**Data Availability Statement:** Not applicable.

**Conflicts of Interest:** The author declares no conflict of interest.

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
