# Peer review of "From Cyberpunk to Cramped Dweller: The Peculiar History of Hong Kong ‘Heterotopias’"

_2409-9252, doi:10.3390/histories1030019_

Round 1

Reviewer 1 Report

This paper looks at three types of architectural structures in Hong Kong--the Kawloon Walled City, the Chungking Mansions, and Cage homes--as they act as metaphors for the history of Hong Kong. Kawloon Walled City suggests the tension of the British/Chinese split within the city and the strategic position that Hong Kong had for China, esp., which coveted it. The Mansions point up the reputation of Hong Kong as a zone for 'vice,' a reputation often looked at hypocritically but that nevertheless represented some aspects of Hong Kong's Westernization that, again, China coveted. Finally, the discussion of slums, cage living, squatters illustrates the very different way of living that Hong Kong demands of some of its residents--the restricted geographic space of Hong Kong pressing some residents into amazingly small, unhealthy living spaces. 

The strength of the paper is in what it adds to knowledge about the history of Hong Kong, esp. what the author calls the "heterotopias" of the city--those places that have been 'othered' or made abject. The peculiar logic of Hong Kong, caught between two different worlds, continents, cultures, is made clearer by this discussion. From a theoretical or methodological stand point the essay works best as straight history. This is especially true when we get to the long concluding section on slums and cage housing. The first two sections--on  the walled city and the mansions--are more "interdisciplinary," as the author calls it, than the last. These first two sections function more as metaphors for Hong Kong and are the richer parts of the essay for that. For example, the Disney-like recreation of "Old China" that the walled city becomes. If I could make any overall suggestion about the essay as a whole it would be to heighten and extend those sections that are more than history and that show how these architectural areas shed light on Hong Kong in their unusual ways.

Some specific suggestions: 1. reconsider the concluding paragraph in light of recent events having to do with the (apparently) iron clad lockdown on Hong Kong. Its freedom, it seems, is now gone or at least changed forever. 2. some grammar suggestions: don't begin sentences with numerals (pages 3 and 12). I don't understand putting "[sic]" after British spelling. The author isn't consistent and these are not mistakes. On page 5, the second sentence should read "In 2005, The Economist named...." On the same page, I can't make sense of the last sentence: "...in its projection marked by ...." Likewise, the first sentence of the second paragraph on page 24.

This is a well-written essay, meticulously researched, and original. I wish only that it was more Foucauldian, but that is a lot to ask of anyone.

Author Response

Firstly, thank you so much for both your helpful critiques/suggestions and compliments of my paper. This would be my first published paper (hopefully) as I am presently a doctoral candidate in my history program. I have applied your suggested revisions to my paper as best as I could. Below is my list of revisions applied to Reviewer 1's comments, as well as a few general revisions applied across the manuscript:

  1. I converted numerals to written form at the beginning of paragraphs on pages 3 and 12, as you specified.
  2. I removed the "[sic]" after British spellings on pages 8, 11, 13, 24, 26, 29, and 35.
  3. Page 5: Regarding the fifth sentence on that page, the sentence now reads: "In 2005, The Economist named..." You recommended this sensical switch of wording.
  4. The last sentence of the first paragraph on page 5 now reads: "This streak of independence fosters an image the world has come to harbor of Hong Kong as a minuscule locality projecting world-renowned distinction and individuality."
  5. The first sentence of the second paragraph (i.e., the topic sentence) on page 24 now reads: "Chungking Mansions symbolizes vivacious globalization, and its inhabitants encounter a slew of new experiences there." You had alluded in your commentary that the previous topic sentence was difficult to understand, and I see your point.
  6. I expanded the paper's concluding paragraph to illuminate recent events in Hong Kong as per your recommendations. That recommendation, in particular, really helped since I wrote the primary draft of this paper in autumn 2019. Specifically, I included the passage of the controversial 2020 security law and its effect on Hongkongers.
  7. General revision: I applied a page break between the conclusion paragraph and the bibliography (pgs. 40-41).
  8. General revision: I reduced the indentation length between the footnote # and the beginning of the actual footnote citation. This revision begins on page 2, lasting until page 38.  

Reviewer 2 Report

This article uses Foucault's 'heterotopia' as a keyword to show readers the history of Hong Kong's past and present urban 'alternative landmarks': the Kowloon Walled City, Chungking Mansions, cage homes and public housing in Hong Kong. As an academic article, I think it touches on a very important and representative spatial form in the history of Hong Kong's urban development. Unfortunately, I think the author's description is limited to listing and juxtaposing these spatial forms/landmarks, without specifying how they relate to each other and how these forms relate to the lives of Hong Kong residents, the social and political state of Hong Kong, the historical discourse of Hong Kong, or even their cultural and visual representations that shaped them. I have the following comments to make about this paper. 1. Literature review The article covers a wide range of urban forms of Hong Kong since the end of WWII. However, the references the author cited are limited. For similar topics, the author can refer to: Rooney, Nuala, 2003. At Home with Density, Hong Kong: Hong Kong University Press. Shelton, Barrie, Karakiewicz, Justyna & Kvan, Thomas, 2011. The Making of Hong Kong, London: Routledge. Frampton, A., Solomon, J.D. & Wong, C., 2012. Cities without ground : a Hong Kong guidebook, Novato, Calif.]: ORO. Christ, E. et al., 2010. Hong Kong typology : an architectural research on Hong Kong building types : in the form of a typology index with short descriptive texts, plans and axonometric diagrams : introduced by a photographic essay and an historical insight, Zurich: GTA. Meanwhile, there are also references in Chinese, for example: 薛求理., 2014. 城境 : 香港建築, 1946-2011 = Contextualizing modernity : Hong Kong architecture 1946-2011. 第1版.., 香港: 商務印書館(香港)有限公司. 2. Methodology Using Foucault’s heterotopia as a framework of talking about urban conditions is not a new idea though it could still be a good fit if used appropriately. I am not very convinced by the author’s usage, as he doesn’t explain clearly how the idea of heterotopia can help us to understand all the urban forms in question better. In my opinion, Foucault explores six types of borders between spaces (public and private, life and death, reality and simulation, secular world and archive, etc.) and heterotopia are spaces of others in each of such pairs of spaces. I agree that Kowloon Walled City, Chungking Mansions and cages homes can be well considered as a kind of Other space, but the author doesn’t state clear, for what spaces they are the Other. I think this problem lies ultimately in the fact that this article is written in a depoliticized manner so that we can’t see the politics between the forces at play that formed those spaces. 3. Structure and organization I feel the structure of the article a bit loose and I am not able to follow the organization of the subtitles in a consistent logic. The article starts with an irrelevant and clichéd summary of the 2019 social movement to which the rest of the paper cannot be related to (though I am not saying it really can’t!). The historical discussion on the public housing comes after two very specific case studies and go back to another specific case study of cage home. For readers, these sub-parts read isolated from each other and every time a new topic is introduced, we have to go back to an earlier period of Hong Kong urban history once again. In sum, the connections between the cases and how they jointly speaks to the idea of heterotopia (and for what purpose this idea is used) are unclear and weak. The paper shows there is no in-depth historical research into the history of how Hong Kong space has been, in Henri Lefebvre’s term, produced.

Reviewer 3 Report

Thanks for submitting this paper. I enjoyed reading your paper. While this paper addresses some housing issues in Hong Kong, I spotted some problems:

  1. You have started with your introduction with a political event and ended with the same political event. What is the purpose of citing this event? Do you want to connect housing issues with politics? But I cannot see you are really doing that in your paper.
  2. You used three examples in your paper: Kowloon Walled City, Chungking Mansions, and cage homes. That is fine. But why in this main text, you suddenly discuss the Hong Kong Housing Authority after Chungking Mansions? My feeling on this paper is you cannot tightly connect the contents of three and bring out the main idea of the paper. It sounds like you are just writing three subjects and try to mix them together but in vain.
  3. You said: ‘This paper, through a critical lens of history, aims to tether much of Hong Kong’s modern strife to deeper historical roots representative of the effects of colonial governance, fundamentally flawed urban planning, and citizen-led self-determination.’ But I cannot see you can really address this.
  4. Quotations with more than 40 words should be block quoted (example: ref 22)
  5. The Governor-General of Kwantung and Kwangsi (Should be Viceroy of the Two Guangs)
  6. You should add the Chinese names for non-English terms right after. For example: Viceroy of the Two Guangs (兩廣總督); Tsungli Yamen(總理衙門); T’an Chung-lin(譚鍾麟)
  7. Page 12 – “prolong the ‘moral danger’ that the City’s notorious suburbs represented and ‘could not fail to exercise a malign influence on the minds of the natives.’”— reference is missing. Check your paper; every quote should have a reference. I spotted a lot of missing references.
  8. Page 13: the Walled City Order in Council should be italicised.
  9. Page 16: what is Ging Yi?
  10. Page 17: Glad to see you citing Gordon Jones, he was my father’s colleague in Hong Kong Government. I think you should read Gavin Ure’s Governors, Politics and the Colonial Office as well. It covers housing policy before the 1960s.
  11. Chungking Mansions: most of the references are from Gordon Mathews’ (麥高登) two books. I know Gordon, and he is an expert on Chungking Mansions. However, over-citing his works in your paper is not a good sign, and you should find other materials whenever possible.
  12. Page 23: the paragraph is too long – consider cutting it into two or three.
  13. Page 25: In the United Kingdom, Crown officials are defined by law, and they are not the same as colonial officials. Do you really mean Crown officials?
  14. Page 26: citations are needed for the paragraph. Especially after EVERY SINGLE DIRECT QUOTATION!
  15. Hong Kong housing plan: you should be aware that Governor Mark Young was planning to launch public housing scheme in 1935, but it was forced to stop due to the Second World War. (can check Ure’s book)
  16. It would help if you discussed the large Chinese influx caused by the Chinese Civil War and the rise of Red China (1949 to the Cultural Revolution). The unexpected refugees made the Hong Kong Government’s Housing Plan unmanageable. The colonial government had already tried to address the housing problem from the post-war to 1997. Remember, most Chinese refugees were squatters living in New Kowloon and were not British subjects, so the government had no obligation to deal with their housing problems. If there was no Red China, the housing plan suggested by Governor Sir Mark Young should be able to address the housing issues of Hong Kong Chinese after the war.
  17. You should be aware that as early as 1949, Governor Sir Alexandar Grantham was concerned with introducing subsidised housing as early as 1949, but encountered opposition from Chinese members of the Legislative Council. If we look back to Sir Mark Young and Sir Alexandar Grantham, then we should know the role of the fire in the history of public housing in Hong Kong has been overstating.
  18. Page 29: The Crown government -> the colonial government.
  19. Page 32: From 1954 until 1984, the government-induced Hong Kong Housing Authority engaged in a “resettlement” program for its dispossessed squatter and tenement dweller demographics. – Why it stopped in 1984?
  20. For the housing authority, you should look at the work of governor Murray MacLehose. For details, you should read Yu, A. C. K. (2020). Was governor MacLehose a great architect of modern Hong Kong?. Asian Affairs51(3), 485-509.
  21. For social injustice in post-1997 Hong Kong you can read Purbrick, M. (2020). Hong Kong: the torn city. Asian Affairs, 51(3), 463-484.
  22. You should also read the late Professor Leo Goodstadt’s papers and books. He wrote many on housing in Hong Kong.
  23. Current housing problems are largely caused by the collision of Chinese land developers (particularly red capitals) and the SAR (and Chinese government). You should discuss this as well. Again Goodstadt provided a lot of discussions which you can look for. You can read Yip, M. (2018). New Town Planning as Diplomatic Planning: Scalar Politics, British–Chinese Relations, and Hong Kong. Journal of Urban History, 0096144220948813.
  24. Bibliography: only listed what you have cited. Check with the journal requirement. You should make a reference list rather than a bibliography.

Author Response

Thank you so much for taking the time to review my essay. Here is a point-by-point response to the reviewer's comments in numerical order:

  1. I do not intend to connect Hong Kong's (HK) housing issues with its politics. I aimed for my introduction to situate the reader within the present state of affairs in HK to allow for easier wading into the deep end of my historical narrative. One of the main issues driving HK's protests was housing, which I mention at the end of the introduction, and use as a springboard to transition the reader into the centuries-long, colonial context of HK's housing evolution and its corresponding inconsistencies and inequalities.
  2. Discussing the Housing Authority is critical because it provides reasons and mechanisms behind the decline of squatter settlements and the rise of other manifestations of housing inequality, namely cage homes, as space is limited under Housing Authority dictates. I cannot discuss the rise of cage homes, for instance, without first discussing the Housing Authority's roles in shaping HK's present housing woes.
  3. The "deeper historical roots" are strung together throughout the narrative, but especially with the "British Hong Kong vs. Chinese Hong Kong" section intended to gauge and paint a picture of how colonial governance shaped the look and distribution of housing HK's many demographics under British rule. The "citizen-led self-determination" reference is a homage to the multifaceted degrees of alegal autonomy that KWC embodied.
  4. I use Chicago Style Block Formatting, which states a block quote is used for 5 or more lines of prose. I double-checked my quotes and their corresponding lengths.
  5. I changed the "Governor-General of Kwantung and Kwangsi" title to "Viceroy of the Two Guangs" on page 10.
  6. I added the Chinese names immediately after non-English terms.
  7. Missing references detected across the paper have been amended.
  8. On page 13, the "Walled City Order in Council" is now italicized.
  9. On page 16, Ging Yi is a Triad gang, as I describe in a revised sentence.
  10. Thank you for your recommendation to read the Ure book that you previously mentioned.
  11. While I agree with you that overciting one author's works does not appear as well-rounded, I made solid attempts to locate other works/authors of comparable quality concerning Chungking Mansions' research. However, Gordon Mathews is truly the only authority with a visible breadth of work on the subject. He led the charge on analyzing the Mansions. Trust me when I say I combed for other sources from other authors.
  12. On page 23, a new paragraph begins with the sentence reading, "Fires were a constant at Chungking."
  13. On page 25, thanks for raising that discrepancy. I meant "colonial officials" and made the necessary adjustment.
  14. I added citations to every direct quotation in the previously mentioned paragraph.
  15. Thank you for supplying the information concerning Gov. Mark Young's "planning to launch public housing in 1935." It will be useful in future research.
  16. Thank you for all those recommendations and for raising those points. However, I purposely chose to omit the Chinese Civil War and the rise of Red China (1949 to the Cultural Revolution) in my scope to keep the spotlight on Hong Kong. While it would have been contextually beneficial and striking to the narrative to include the mainland to a greater degree into my narrative/research, I had to cap this essay at some point. The rise of Red China and its influence on housing plans suggested by Gov. Sir Mark Young would make for a captivating follow-up essay related to this essay.
  17. Thank you for raising those points regarding Sir Grantham and Sir Young.
  18. "Crown" replaces "colonial" on pages 8, 23, and 28.
  19. Your query related to 1984 on page 32 is answered in brief.
  20. Thank you for illuminating the work of Gov. Murray MacLehose and the Yu piece.
  21. Thank you for highlighting the social injustice piece by Purbrick (2020).
  22. Thank you for your suggestion on reading Prof. Goodstadt's papers and books.
  23. Thank you for the suggestion, but I chose not to incorporate a mainland "red capital" influence not to appear having the essay meander in too many directions. I wanted to keep the paper, especially the first half, exclusively rooted in the British colonial era.
  24. According to the Chicago/Turabian style, which historians use, a bibliography is used when incorporating Chicago footnote citations in the text. This is also evident by where I place the publication year near the end of an entry (bibliographic form). I did, however, significantly revamp the bibliography. I omitted text types (newspapers, articles, books, etc.) and source types (primary, secondary). I condensed them into one flowing bibliography with all newspaper sources included in alphabetical order of newspaper name, article title, and corresponding author(s).   

Round 2

Reviewer 2 Report

-

Author Response

Thank you for taking the time to review my manuscript.

Reviewer 3 Report

Thanks for making a revised version

I guess you are obviously a PhD student. 

You don't need to mention the Chinese name all the time. You just need to mention in the first appearance, then hereafter you can just just the English name. For example, you mentioned the Chinese name of the Viceroy of the Two Guangs (兩廣總督) on page 10, then you don't need to repeat the same as the readers will know. 

Referencing style: I am not sure if the editor will check again. But you should follow the JOURNAL'S POLICY instead of your field practice. (I am a historian from the UK and we use MHRA in the UK). So check with the policy of the journal that you are submitting, not your field practice. 

It would be great of you can consider the works I mentioned in the first review. If will help to improve your work and readability. 

Author Response

Thank you for this latest round of constructive feedback.

I will make the appropriate changes in reference to listing the Chinese names.

I will check the referencing style again just to be sure.

I will submit a revised draft in about a month once I comb through the literature you recommended and see if its inclusion is applicable to my essay's themes and direction. You will receive it when that time comes.

Thanks again for all your critiques thus far.

Round 3

Reviewer 3 Report

I am happy to accept the paper. Good luck in your publishing journey.